# HPV Vaccine Hesitancy and Influencing Factors among University Students in China: A Cross-Sectional Survey Based on the 3Cs Model

**DOI:** 10.3390/ijerph192114025

**Published:** 2022-10-28

**Authors:** Yan Huang, Cheng Chen, Lei Wang, Huamei Wu, Ting Chen, Luying Zhang

**Affiliations:** 1Center for Chinese Public Administration Research, School of Government, Sun Yat-sen University, Guangzhou 510275, China; 2School of Public Health, Wuhan University of Science and Technology, Wuhan 430065, China; 3National Institute of Hospital Administration, National Health Commission, Beijing 100044, China; 4School of Public Administration and Emergency Management, Institute of Public Policy, Jinan University, Guangzhou 510632, China; 5Shanghai Urban Construction Vocational College, Shanghai 200438, China; 6School of Public Health, Fudan University, Shanghai 200032, China

**Keywords:** HPV vaccine, hesitancy, influencing factors, 3Cs model, university students

## Abstract

The burden of disease caused by cervical cancer ranked second among female tumors in China. The HPV vaccine has been proven to be a cost-effective measure to prevent cervical cancer, but the vaccination rate remained low to date among university students. This study aimed to understand the status quo of HPV vaccine hesitancy among university students across China during the COVID-19 pandemics and systematically analyze determinants of HPV vaccine hesitancy based on the WHO 3Cs model. Cross-sectional data were collected using an online survey of female university students in four cities across China in June 2022. Multinomial logistic regression was adopted to determine factors influencing vaccine hesitancy based on the 3Cs model with three dimensions, namely complacency, convenience, and confidence. Among 1438 female university students surveyed in this study, 89.7% did not hesitate to vaccinate against HPV, only 8.9% hesitated to some extent, and 1.4% refused to vaccinate. The actual vaccination rate for the HPV vaccine was 34.2%. Based on the 3Cs model, this study found that the trust on the efficacy of vaccines, risk perception of being infected by HPV, price, and distance/time were influencing factors of vaccine hesitancy. Knowledge of the HPV vaccine and sociodemographic characteristics, such as education levels, were also statistically relevant. Therefore, it is recommended that relevant scientific knowledge on cervical cancer and the HPV vaccine should be spread on campus, the vaccination appointment procedure should be simplified, and the affordability of vaccination should be increased through strategic purchasing or providing subsidies, so as to reduce HPV vaccine hesitancy.

## 1. Introduction

Cervical cancer is one of the most common malignancies among females worldwide, with persistent human papillomavirus (HPV) infection as its necessary cause [1]. In China, cervical cancer incidence rate (17.69/105) and mortality rate (5.52/105) ranked second among female tumors [2], and there were 110 thousand new cases and 59,060 deaths of cervical cancer in 2020 [3], bringing about substantial economic and social burden [2,3].

The HPV vaccine has been proved to be a cost-effective measure to prevent cervical cancer around the world [4] and is highly recommended by the World Health Organization (WHO) [5]. In China, the HPV vaccine was approved for listing in 2016 as a self-paid vaccine; however, the vaccination rate remained low to date. For female university students more specifically, who were the target population, the reported HPV vaccination rate ranged from 3.3% to 14.09% [6,7,8], which was much lower than that of similar population in both developed and developing countries, such as Australia (56%) [9], Switzerland (69%) [10], and India (21%) [11].

Vaccine hesitancy, which refers to a delay in the acceptance or refusal of vaccines when available [12], has dampened public enthusiasm for vaccination, and has been listed by the WHO as one of the top 10 threats to global health in 2019 [13]. A decline in the HPV vaccination rate due to HPV vaccine hesitancy has been reported in many countries in recent years, such as in Japan [14], Brazil [15], India [11], and Jordan [16]. Therefore, exploring the factors influencing HPV vaccine hesitancy can help to reduce hesitancy and increase vaccination acceptance.

A number of previous studies were conducted to understand attitudes towards HPV vaccines among university students in China, which mainly focused on the awareness of cervical cancer and the HPV vaccine, and vaccination willingness [6,7,8] [17,18,19,20]. However, only few studies measured HPV vaccine hesitancy directly, with the proportion among female university students reaching 37.57% [21]. The proportion of university students who were unwilling to receive the HPV vaccine, though not equal to vaccine hesitancy, varied from 11.55% to 43.1% [6,7,8,17,18,19,20], as those studies previously discussed were mostly conducted in a single city with a small population [6,7,8,17,18,19,20,21] or with a special group of population [22].

To analyze the influencing factors of vaccine hesitancy, the SAGE of the WHO proposed the 3Cs model to comprehensively analyze determinants of vaccine hesitancy from three dimensions, namely complacency, convenience, and confidence [23]. This 3Cs model has been widely adopted by researchers to analyze COVID-19 and influenza vaccine hesitancy [24,25]. However, it was less frequently used on HPV vaccine hesitancy in previous studies in China, as most previous studies did not adopt an explicit framework in analyzing determinants [6,7,8,17,18,19,20]. Therefore, there is a lack of empirical evidence on HPV vaccine hesitancy and its determinants among university students from multi-center large-sample-size investigations.

This study aimed to understand the status quo of HPV vaccine hesitancy among female university students across China during the COVID-19 pandemics and systematically analyze determinants of HPV vaccine hesitancy based on the WHO 3Cs model. The hypothesis was that 3C factors concerning complacency, confidence, and convenience were associated with HPV vaccine hesitancy.

## 2. Methods

### 2.1. Study Setting

The first HPV vaccine listed in the Chinese market was approved by the National Medical Products Administration (NMPA) in July 2016 as the self-paid bivalent HPV vaccine for women aged 9–45 [26]. Subsequently, the quadrivalent HPV vaccines for those aged 20–45 and 9-valent HPV vaccines for women aged 16–26 were approved [27]. The NMPA announced that the first homegrown HPV vaccine in China was approved for marketing on 31 December 2019. Later, the NMPA reduced the minimum age for quadrivalent vaccines to 9 years in November 2020 [28]. In addition, in order to effectively elevate vaccination rate, a growing number of cities in China provided domestic bivalent HPV vaccines to teenage groups for free.

### 2.2. Study Design

This study used a multicenter cross-sectional and survey-based research methodology. The independent variables in the study were constructs of the “3Cs” model, while the dependent variable was the intention/hesitancy for taking HPV vaccination. Participants were included in this study if they: (1) were currently enrolled as an undergraduate or graduate student at a sampling university; (2) were 18 years old or older; (3) were female; (4) had access to the Internet via computer or smart phone; and (5) provided informed consent. This study was ethically reviewed and approved by the Institutional Review Board, School of Public Health, Fudan University (IRB#2022-08-0992).

### 2.3. Survey Instruments

The survey instrument, developed by the research group, consisted of items concerning HPV vaccine hesitancy, determinants from the vaccine hesitancy “3Cs” model, knowledge of the HPV vaccine, and socio-demographic characteristics. The question to measure vaccine hesitancy, “if the HPV vaccine was offered to you today, do you have any intention in taking it?”, could be answered with one of the following responses: “acceptance without hesitancy”, “hesitancy”, and “refusal”. Eleven questions concerning “3Cs” factors were developed to measure the participants’ attitudes on complacency, as well as their convenience and confidence (Table 1). Two questions about their knowledge of HPV and the HPV vaccine were proposed to examine participants’ cognition. Socio-demographic items were developed to collect individual information on age, major, education level, health insurance, living expenses, family history of cancer, previous vaccination behaviors and so on. Personal identifiers (i.e., names, emails, and phone numbers) were not collected during survey. A pilot survey, which was conducted with 60 college students to test the readability and comprehension of the questionnaire, was found to be easily understood. The instrument was administered via Wenjuanxing, an online survey tool.

### 2.4. Sample Size and Data Collection

This study used a stratified sampling method. After comprehensively considering geographical location and socioeconomic development level in China, four cities at different regions were selected as sampling cities, including Shanghai, Guangzhou, Wuhan, and Nanning, with the former as Municipal City and the last three as the capital cities of provinces located in eastern, central, and western regions, respectively. In each city, four to five universities were chosen to cover students with a different education level (college, undergraduate, or postgraduate) and majors (science and engineering, social science, or medicine).

A priori sample size was calculated with the following formula based on an error α of 0.05 and a permissible error δ of 0.05:(1)$$n=\frac{Z_{1-\alpha /2}^{2}\times p\left(1-p\right)}{\delta^{2}}$$*n* represents the sample size required for the survey, $Z_{1-\alpha /2}^{2}$ is the standard normal deviation of α ($Z_{1-\alpha /2}^{2}$ = 1.96 when α error of 0.05), and p is the expected prevalence or positive rate of the survey. According to two recent meta-analysis of 31 studies, the acceptance rates of the HPV vaccine among university students in mainland China were 68.0% [29] and 71.8% [30], respectively. Based on that, this study conservatively assumed that the HPV vaccine acceptance rate was 70.0%. Substituting *p* = 0.70 into the formula, n would be 322 in each city according to various calculations. Thus, the total sample size of this study was expected to be 1288 female university students for four cities.

Data were collected between 1 June and 30 June 2022 using the Wenjuanxing online survey tool. Students were recruited through advertisements in class announcement. The landing page of the electronic survey instrument provided informed consent and study information. Participants could start to fill out the questionnaire only if they clicked the button which represented that they consented to all information provided and were willing to participate in the study. As for quality control, online questionnaires finished in no less than 180 s with all required items filled were considered as valid. In total, 1498 online questionnaires were collected, and 1438 questionnaires were valid through quality checks.

### 2.5. Statistical Analysis

The data from online questionnaires were downloaded as a Microsoft Excel file and then imported into IBM SPSS version 26.0 (IBM Corp., Armonk, NY, USA) for data analyses. Chi-square tests and multiple logistic regression were used to assess the associations between sociodemographic factors, 3Cs factors, and vaccine hesitancy. The chi-square test was first used to determine factors associated with vaccine hesitancy. Multinomial logistic regression was then adopted to assess the correlations between independent variables and college students’ attitudes toward HPV vaccination (“acceptance with no hesitancy” as the reference group; “hesitancy” and “refusal” as the comparison groups). The results of multinominal logistic regression were shown in the form of odds ratios (ORs) and 95% confidence intervals (CIs). A *p* value < 0.05 was considered statistically significant.

## 3. Results

### 3.1. Demographic Characteristics and Knowledge of HPV Vaccines

A total of 1438 valid questionnaires were collected in this study. As shown in Table 2, the proportions of female university students aged <20, 20–26, and >26 years old were 25.0%, 72.2%, and 2.9%, respectively. Among them, 67.5% were undergraduate students. Overall, 45.8% and 43.7% of participants majored in medicine and social science, respectively. Moreover, 63.8% had a monthly living expenses consumption between CNY 1001 and 2000 (USD 144.5–288.7). In addition, 47.1% of students’ families lived in urban areas, 93.9% were enrolled in basic medical insurance, 91.2% had no family cancer history, and 75.6% of people did not receive the vaccine at their own expense over the past five years. As for the knowledge of cervical cancer and the HPV vaccine, 67.6% and 21.7% had a moderate and low score, respectively. Furthermore, 64.0% of university students did not receive HPV vaccine recommendations by medical staff, and 83.2% were recommended the HPV vaccine by people around them.

### 3.2. Vaccine Hesitancy Level

Among all female university students, 1290 students (89.7%) were willing to take up the HPV vaccine without hesitancy, 128 (8.9%) were hesitant, and only 20 (1.4%) refused to receive the HPV vaccine. Among those who were willing, 492 students were already vaccinated or already made appointments to become vaccinated, meaning that the total vaccination rate was 34.2%. Table 2 also shows the associations between vaccine hesitancy and sociodemographic characteristics. Sociodemographic characteristics, such as city, education level, having a family history of cancer or not, having received and paid for the vaccine over the past five years or not, the level of HPV understanding, and having been recommended the HPV vaccine by people around and medical staff or not, were significantly related to vaccine hesitancy (*p* < 0.05).

### 3.3. Perceptions and Attitudes toward HPV Vaccination

Table 3 shows the distribution of 3Cs factors and their associations with vaccine hesitancy. Most university students thought that the cervical cancer was very harmful to health (68.8%) and that HPV vaccination was necessary (89.9%). In total, 36.5% of students had a greater fear of cervical cancer and 17.20% thought that they were more likely to be infected by HPV. Only a few said that the price (6.10%) or time/distance (4.40%) hindered HPV vaccination (Table 3). Additionally, 71.70% of students did not hear any vaccine-related negative information, and most students had trust in the safety (89.0%) and efficacy (88.0%) of HPV vaccines and advice from medical staff (80.4%).

Most of the above characteristics were significantly related to vaccine hesitancy (*p* < 0.05). It was revealed that higher perceived susceptibility and harm associated with HPV, a higher perceived importance of HPV vaccination, and recommendations from medical staff were significantly related to lower vaccine hesitancy. Trust in the advice of medical staff, and the safety and efficacy of HPV vaccination also became related factors to lower vaccine hesitancy. In the meanwhile, price or distance/time significantly hindered vaccination. However, the impact and relevance of hearing negative vaccine-related information was not significant in our study.

### 3.4. Influencing Factors of HPV Hesitancy

The results of multinomial logistic regression for the influencing factors of HPV hesitancy based on the 3Cs model is shown in Table 4. Students who perceived high efficacy in the HPV vaccination were less likely to be hesitant towards vaccination (trust: OR = 0.084, CI: 0.014–0.494; neutral: OR = 0.112, CI: 0.021–0.583). Students who perceived HPV infection severity to be low were more likely to be hesitant towards vaccination (OR = 4.007, CI: 1.584–10.138). Students who perceived the vaccine price to be acceptable were less likely to be hesitant towards vaccination (OR = 0.128, CI: 0.055–0.299), as were those who considered the inconvenience of time/distance (OR = 15.366, CI: 6.702–35.229).

Besides that, students with more knowledge of HPV and the HPV vaccine were less likely to have vaccine hesitancy (high scores, OR = 0.140, CI: 0.023–0.845; moderate scores, OR = 0.443, CI: 0.236–0.833). The possibility of HPV vaccine hesitancy was higher for undergraduate students (OR = 3.407, CI: 1.504–7.717) and junior college students (OR = 2.800, CI: 1.130–6.940).

## 4. Discussion

Based on the WHO vaccine hesitancy 3Cs model, this study comprehensively examined the level of HPV vaccine hesitancy among university students in four cities across China, and systematically investigated its influencing factors.

Among the female university students surveyed in this study, 89.7% did not hesitate to receive the vaccine against HPV, only 8.9% hesitated to some extent, and 1.4% refused to vaccinate. Compared with previous studies on HPV vaccine hesitancy or unwillingness to vaccinate among Chinese university students (11.6% to 43.1%) [6,7,8,17,18,19,20,21,29,30] and adult women in other countries (25.0% to 33.8%) [16,31], this study reported a relatively lower level of vaccine hesitancy and refusal, as well as a higher level of HPV vaccination acceptance. At the same time, this study found that 34.2% of female students were already vaccinated or already made appointments to receive the HPV vaccine, which was also higher than the results in previous studies [6,7,8]. Considering that this study is the first multicenter survey to investigate the hesitancy of Chinese female university students to receive the HPV vaccine after the COVID-19 pandemic, people’s risk perception and vaccination willingness might be affected to some extent [32,33,34,35].

Our study found that self-confidence, complacency, and convenience have a particular influence on HPV vaccine hesitation among Chinese university students.

In the confidence dimension, we found that university students who trusted the advice of medical staff and believed that the HPV vaccine was safe and effective were less likely to hesitate to receive the HPV vaccine. Previous studies have revealed that confidence is the most relevant factor for vaccine hesitancy, and safety and effectiveness were the two most important factors for deciding to receive vaccination [36,37]. It is worth noting that, compared with previous research results [38], this study found that there was no significant difference in the impact of hearing about negative vaccine-related news on vaccine hesitation. The recommendations and suggestions of medical personnel on vaccines were important sources of information for most people (80.4%). Therefore, medical personnel can spread relevant knowledge and provide suggestions during diagnosis and treatment or on other occasions, which can help vaccinators correctly understand the harm of cervical cancer and the benefits of HPV vaccination [31].

In the complacency dimension, risk perception also had a significant influence on HPV vaccine hesitancy. University students who were more afraid of cervical cancer, conceived that they were more likely to be infected by HPV, and considered that cervical cancer was more harmful often did not hesitate to vaccinate, which was consistent with previous studies [17]. Especially under the COVID-19 epidemic circumstances, the public’s risk perception of potentially high-incidence disease may be stronger than before. Therefore, we need to provide cervical cancer and HPV vaccine knowledge for university students to improve their awareness of the HPV vaccine in order to reduce their hesitancy.

In the convenience dimension, price and time (or distance) were obstacles to HPV vaccination among university students, which were significant in the chi-square test and multinominal logistic regression. As a growing number of cities in China started to offer the two-valent HPV domestic vaccine free for charge for teenagers, the price obstacle for HPV vaccination may be reduced to some extent. In addition, a complicated appointment process was also a disturbing factor affecting HPV vaccination, highlighting the need to simplify the HPV vaccination appointment process according to public demand.

Moreover, it is worth mentioning that there is still a gap between the willingness to vaccinate and actual vaccination behavior. The existing literature shows that the willing acceptance rate of vaccination was higher than the actual vaccination rate [6,8,11,16,21,31], and that the two are positively correlated. The conversion of willingness into action will be affected by factors such as a perceived low risk to develop HPV infection, a lack of information and knowledge of HPV, concerns about the effectiveness and safety of the vaccines, and concerns about the high cost of vaccines [6,8,11,16,21,31]. This study focuses on exploring the influencing factors of vaccine hesitancy, so as to improve vaccination intention. The facilitating factors that motivate the transformation of vaccination willingness into actual vaccination behavior need to be further explored in future research.

To our knowledge, this is the first study to comprehensively examine the vaccine hesitancy and influencing factors based on the 3Cs model among university students across China. By adopting a stratified sampling method to select sample from four cities with different socio-economic development levels, this study has a sufficient sample size and good representativeness. It can provide useful information to help us understand university students’ HPV vaccine hesitancy and develop vaccination strategies under the COVID-19 pandemic circumstances.

This study also has some limitations. First, the online survey may have selection bias in the sense that only university students who were interested in the HPV vaccine were investigated. Second, with a cross-sectional design, this study could only reveal the correlation between influencing factors and HPV vaccine hesitancy rather than causal relationships. Third, for investigation convenience, cities located in North China were not included in our sample cities, which may influence the generalization of our results to some extent.

## 5. Conclusions

The study examined current HPV vaccine hesitancy among female university students in China, revealing that around 89.7% of students were willing to receive the HPV vaccine without hesitancy and that 8.9% had vaccine hesitancy. Based on WHO’s vaccine hesitancy 3Cs model, this study found that trust in relation to the efficacy of vaccines, the risk perception of being infected by HPV, price, and distance/time were associating factors which impacted vaccine hesitancy. HPV knowledge and education levels were also statistically relevant. It is recommended that relevant scientific knowledge of cervical cancer and the HPV vaccine should be spread on campus, that the vaccination appointment procedure should be simplified, and that the affordability of vaccination should be increased through strategic purchasing or providing subsidies, so as to reduce HPV vaccine hesitancy.

## Figures and Tables

**Table 1 ijerph-19-14025-t001:** Questions used to measure “3Cs” factors in survey instrument.

Factors	Questions	Question Design
Complacency	What do you think of the severity of being infected by HPV?	Five-point Likert scale (“very low” to “very high”)
	What do you think of the probability of being infected by HPV?	
	How do you fear of being infected by HPV?	
	What do you think of the necessity of HPV vaccination?	
Confidence	What do you think of the efficacy of vaccines?	
	What do you think of the safety of domestic vaccines?	
	What do you think of the safety of vaccines abroad?	
	Do you agree that the vaccine-related advice provided by medical staffs is reliable?	
	Have you heard the negative information about vaccines?	Binary (“yes” or “no”)
Convenience	Do you agree that the price prevents you from vaccinating against HPV?	
	Do you agree that time/distance prevents you from vaccinating against HPV?	

**Table 2 ijerph-19-14025-t002:** Characteristics and HPV knowledge of university students by hesitancy status.

Factors		Total, *n* (%)	The Attitude to HPV Vaccination, *n* (%)		*p*-Value
			No Hesitancy (*n* = 1290)	Hesitancy (*n* = 129)	
Age (years)	<20	359 (25.0)	326 (90.8)	29 (8.1)	0.365
	20–26	1038 (72.2)	929 (89.5)	95 (9.2)	
	>26	41 (2.9)	35 (85.4)	4 (9.8)	
City	Shanghai	440 (30.6)	392 (89.1)	43 (9.8)	0.038
	Wuhan	349 (24.3)	318 (91.1)	25 (7.2)	
	Guangzhou	329 (22.9)	301 (91.5)	26 (7.9)	
	Nanning	320 (22.3)	279 (87.2)	34 (10.6)	
Education level	Junior college	245 (17.0)	215 (87.8)	28 (11.4)	0.011
	Undergraduate	970 (67.5)	864 (89.1)	93 (9.6)	
	Postgraduate	223 (15.5)	211 (94.6)	7 (3.1)	
Major	Science and technology or agriculture	151 (10.5)	138 (91.4)	11 (7.3)	0.885
	Social sciences	628 (43.7)	558 (88.9)	61 (9.7)	
	Medicine	659 (45.8)	594 (90.1)	56 (8.5)	
Residence	Rural	761 (52.9)	675 (88.7)	74 (9.7)	0.404
	Urban	677 (47.1)	615 (90.8)	54 (8.0)	
Medical insurance	Yes	1350 (93.9)	1211 (89.7)	120 (8.9)	0.977
	No	88 (6.1)	79 (89.8)	8 (9.1)	
Living expense (CNY)	<1000	177 (12.3)	150 (84.7)	22 (12.4)	0.060
	1001–2000	918 (63.8)	830 (90.4)	78 (8.5)	
	2001–3000	283 (19.7)	253 (89.4)	27 (9.5)	
	>3000	60 (4.2)	57 (95.0)	1 (1.7)	
Family history of cancer	Yes	127 (8.8)	116 (91.3)	4 (3.1)	<0.001
	No	1311 (91.2)	1174 (89.5)	124 (9.5)	
Were vaccinated at own expense over the past 5 years	Yes	351 (24.4)	328 (93.4)	20 (5.7)	0.030
	No	1087 (75.6)	962 (88.5)	108 (9.9)	
HPV and vaccine knowledge	High scores (7–9)	154 (10.7)	147 (95.5)	4 (2.6)	<0.001
	Moderate scores (4–6)	972 (67.6)	894 (92.0)	71 (7.3)	
	Low scores (0–3)	312 (21.7)	249 (79.8)	53 (17.0)	
HPV vaccines recommended by people around them	Yes	1196 (83.2)	1095 (91.6)	87 (7.3)	<0.001
	No	242 (16.8)	195 (80.6)	41 (16.9)	
HPV vaccines recommended by medical staff	Yes	518 (36.0)	487 (94.0)	28 (5.4)	<0.001
	No	920 (64.0)	803 (87.3)	100 (10.9)	

Refusal (*n* = 20) was not shown in this table because few or no participants chose this option.

**Table 3 ijerph-19-14025-t003:** Chi-square results of 3Cs factors by hesitancy status.

3C Domains	Factors		Total, *n* (%)	The Attitude to HPV Vaccination, *n* (%)		*p*-Value
				No Hesitancy	Hesitancy	
Complacency	The severity of being infected by HPV	Mild	69 (4.8)	51 (73.9)	12 (17.4)	<0.001
		Moderate	379 (26.4)	331 (87.3)	40 (10.6)	
		Very severe	990 (68.8)	908 (91.7)	76 (7.7)	
	The probability of being infected by HPV	low	452 (31.4)	391 (86.5)	50 (11.1)	0.039
		moderate	739 (51.4)	674 (91.2)	57 (7.7)	
		high	247 (17.2)	225 (91.1)	21 (8.5)	
	The fear of being infected by HPV	Mild	263 (18.3)	223 (84.8)	29 (11.0)	<0.001
		Moderate	650 (45.2)	581 (89.4)	61 (9.4)	
		Very fear	525 (36.5)	486 (92.6)	38 (7.2)	
	The necessity of HPV vaccination	Necessary	1320 (89.9)	1237 (93.7)	81 (6.1)	<0.001
		Neutral	113 (7.7)	52 (46.0)	46 (40.7)	
		Unnecessary	5 (0.4)	1 (20.0)	1 (20.0)	
Confidence	The negative information about vaccines	Yes	407 (28.3)	370 (90.9)	34 (8.4)	0.363
		No	1031 (71.7)	920 (89.2)	94 (9.1)	
	The safety of domestic vaccines	Trust	965 (67.1)	899 (93.2)	60(6.2)	<0.001
		Neutral	435 (30.3)	357 (82.1)	65 (14.9)	
		Distrust	38 (2.6)	34 (89.5)	3 (7.9)	
	The safety of vaccines abroad	Trust	1084(75.4)	1014 (93.5)	64 (5.9)	<0.001
		Neutral	343 (23.9)	267 (77.8)	63 (18.4)	
		Distrust	11 (0.8)	9(81.8)	1 (9.1)	
	The efficacy of vaccines	Trust	1266 (88.0)	1177(93.0)	80 (6.3)	<0.001
		Neutral	163 (11.3)	109 (66.9)	44 (27.0)	
		Distrust	9 (0.6)	4 (44.4)	4 (44.4)	
	The vaccine-related advice provided by medical staffs is reliable	Trust	1156 (80.4)	1085 (93.9)	62 (5.4)	<0.001
		Neutral	272 (18.9)	199 (73.2)	62 (22.8)	
		Distrust	10 (0.7)	6 (60.0)	4 (40.0)	
Convenience *	Price prevents from vaccinating against HPV	No	1350 (93.9)	1290 (95.6)	48 (3.6)	<0.001
		Yes	88 (6.1)	0 (0.0)	80 (90.9)	
	Time/distance prevents from vaccinating	No	1375 (95.6)	1290 (93.8)	67 (4.9)	<0.001
		Yes	63 (4.4)	0 (0.0)	61 (96.8)	

* Fisher’s exact test instead of the chi-square test was used because one of the frequencies was zero. Refusal (*n* = 20) was not shown in this table because few or no participants chose this option.

**Table 4 ijerph-19-14025-t004:** Multinomial logistic regression to identify factors associated with HPV vaccine hesitancy.

	Factors		Hesitancy vs. No Hesitancy			
			OR	*p*-Value	95% CI	
3Cs	The efficacy of vaccines	(Rf: distrust)				
		Trust	0.084	0.006	0.014	0.494
		Neutral	0.112	0.009	0.021	0.583
	The severity of being infected by HPV	(Rf: High)				
		Low	4.007	0.003	1.584	10.138
		Neutral	1.349	0.369	0.702	2.593
	Price prevents from vaccinating against HPV	(Rf: Yes)				
		No	0.128	<0.001	0.055	0.299
	Time/distance prevents you from vaccinating against HPV	(Rf: No)				
		Yes	15.366	<0.001	6.702	35.229
Other	Education Level	(Rf: Postgraduate)				
		Junior college	2.800	0.026	1.130	6.940
		Undergraduate	3.407	0.003	1.504	7.717
	Knowledge of HPV and HPV vaccine	(Rf: Low scores (0–3))				
		High scores (7–9)	0.140	0.032	0.023	0.845
		Moderate scores (4–6)	0.443	0.011	0.236	0.833

Results for the refusal group (*n* = 20) are not shown in the table and can be found in the Appendix A.

## Data Availability

Not applicable.

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
