# Peer review of "HPV Vaccine Hesitancy and Influencing Factors among University Students in China: A Cross-Sectional Survey Based on the 3Cs Model"

_ijerph, 2022, doi:10.3390/ijerph192114025_

Round 1
Reviewer 1 Report
Please see the attached review report.

Reviewer 2 Report
The work entitled:" HPV Vaccine Hesitancy and Influencing Factors Among University Students in China: A Cross-sectional Survey Based on Cs Model" done by Huang etal is not well presented and is of moderate importance. There are some points should be addressed before being published.
1- in the introduction, the research question and the research hypothesis are need to be clarified.
2- In introduction, no information about previous similar works which is so many.
3- In introduction, the author need to clarify the challenges more.
4- In methods, clarify the sample size
5- Little English errors, please go over the manuscript.
Round 2
Reviewer 1 Report
The paper has been improved very well in this revision. The authors have addressed most of my comments. Well done!
One last comment is that, if possible, I hope the authors could explain or discuss that why the survey data shows low rates of HPV vaccine hesitancy in female students in the universities of China but in practice the overall HPV vaccine rate in the target population remains low in China ?
Reviewer 2 Report
It is ok now
Author Response
We are grateful to the reviewer for the careful reading of the paper and the constructive suggestions for improving our work.
Thanks again to the reviewer for the support and affirmation of the article.
All the authors